# Synthesis of Novel Nicotinic Ligands with Multimodal Action: Targeting Acetylcholine α4β2, Dopamine and Serotonin Transporters

**DOI:** 10.3390/molecules24203808

**Published:** 2019-10-22

**Authors:** Juan Pablo González-Gutiérrez, Hernán Armando Pessoa-Mahana, Patricio Ernesto Iturriaga-Vásquez, Miguel Iván Reyes-Parada, Nicolas Esteban Guerra-Díaz, Martin Hodar-Salazar, Franco Viscarra, Pablo Paillali, Gabriel Núñez-Vivanco, Marcos Antonio Lorca-Carvajal, Jaime Mella-Raipán, María Carolina Zúñiga

**Affiliations:** 1Departamento de Química Orgánica y Fisicoquímica, Facultad de Ciencias Químicas y Farmacéuticas, Universidad de Chile, 8380492 Santiago, Chile; jpgonzalezg@ug.uchile.cl (J.P.G.-G.); niko.g.diaz@gmail.com (N.E.G.-D.); 2Departamento de Ciencias Químicas y Recursos Naturales, Facultad de Ingeniería y Ciencias, Universidad de la Frontera, 4811230 Temuco, Chile; martinhodar@gmail.com (M.H.-S.); franco.viscarra@gmail.com (F.V.); p.paillali01@ufromail.cl (P.P.); 3Center of Excellence in Biotechnology Research Applied to the Environment, Universidad de La Frontera, 4811230 Temuco, Chile; 4Centro de Investigación Biomédica y Aplicada (CIBAP), Escuela de Medicina, Facultad de Ciencias Médicas, Universidad de Santiago de Chile, 9170022 Santiago, Chile; miguel.reyes@usach.cl; 5Facultad de Ciencias de la Salud, Universidad Autónoma de Chile, 3467987 Sede Talca, Chile; 6Centro de Bioinformática y Simulación Molecular, Universidad de Talca, 3340000 Talca, Chile; gabioin@gmail.com; 7Escuela de Ingeniería Civil en Bioinformática, Universidad de Talca, Av. Lircay 3340000 Talca, Chile; 8Instituto de Química y Bioquímica, Facultad de Ciencias, Universidad de Valparaíso, 2360102 Valparaíso, Chile; jaime.mella@uv.cl (J.M.-R.); marcos.lorca.c@gmail.com (M.A.L.-C.); 9Departamento de Química Inorgánica and Analítica, Facultad de Ciencias Químicas y Farmacéuticas, Universidad de Chile, 8380492 Santiago, Chile; mczuniga@ciq.uchile.cl

**Keywords:** nAChR, DAT, allosteric modulators, SERT, α4β2, allosteric modulators, affinity

## Abstract

Nicotinic acetylcholine receptors (nAChRs), serotonin transporters (SERT) and dopamine transporters (DAT) represent targets for the development of novel nicotinic derivatives acting as multiligands associated with different health conditions, such as depressive, anxiety and addiction disorders. In the present work, a series of functionalized esters structurally related to acetylcholine and nicotine were synthesized and pharmacologically assayed with respect to these targets. The synthesized compounds were studied in radioligand binding assays at α4β2 nAChR, h-SERT and h-DAT. SERT experiments showed not radioligand [^3^H]-paroxetine displacement, but rather an increase in the radioligand binding percentage at the central binding site was observed. Compound **20** showed K_i_ values of 1.008 ± 0.230 μM for h-DAT and 0.031 ± 0.006 μM for α4β2 nAChR, and [^3^H]-paroxetine binding of 191.50% in h-SERT displacement studies, being the only compound displaying triple affinity. Compound **21** displayed K_i_ values of 0.113 ± 0.037 μM for α4β2 nAChR and 0.075 ± 0.009 μM for h-DAT acting as a dual ligand. Molecular docking studies on homology models of α4β2 nAChR, h-DAT and h-SERT suggested potential interactions among the compounds and agonist binding site at the α4/β2 subunit interfaces of α4β2 nAChR, central binding site of h-DAT and allosteric modulator effect in h-SERT.

## 1. Introduction

Nicotine (Figure 1) is the chemical substance in tobacco responsible for smoking addiction [1], which causes major public health risks. Inhaled cigarette smoke delivers nicotine rapidly to the brain, where it binds to nicotinic acetylcholine receptors (nAChRs) and modulates the release of several neurotransmitters [2]. The effects of nicotine are mainly mediated through specific nAChRs that function as heteropentameric ligand-gated ion channels composed of α4- and β2-subunits [3,4]. These α4β2 nAChRs trigger downstream dopamine signaling in the mesolimbic system [5,6], which is an area in the brain that plays an important role in pleasure and reward sensation [7,8]. The α4β2* nAChRs mediate many behaviors related to nicotine addiction and are the primary targets for currently approved smoking cessation agents [9]. However, the success of these strategies must so far be considered limited, as the only α4β2 ligands currently approved for medical use are the partial α4β2 nAChR agonists Varenicline [10], a partial agonist which binds to α4β2 nAChRs with higher affinity but lower efficacy than nicotine, and Cytisine (Figure 1) [11], both of which are used as a smoking cessation aids [12,13]. Varenicline (Figure 1) elicits a moderate and sustained increase of dopamine levels in the brain reward system [14,15], which would elevate low dopamine levels observed during smoking cessation attempts [16].

A substantial proportion of all smokers have a history of depression, and among people with depression, smoking prevalence is about twice as high as in the general population [17,18]**,** leading to increased morbidity and premature mortality. Earlier research raised concerns that smoking cessation may lead to an increase in symptoms, recurrence or even emergence of depression [19].

Symptoms of depression can be induced in humans through blockade of acetylcholinesterase (AChE) whereas antidepressant-like [20] effects can be produced in animal models and some clinical trials by limiting activity of acetylcholine (ACh, Figure 2) receptors [21]. Thus, ACh signaling could contribute to the etiology of mood dysregulation. Furthermore, a vast number of articles emphasize the significant role of ACh on the initiation and maintenance of drug addiction due to the interactions of the cholinergic system with other neurotransmitter systems, mainly in the ventral tegmental area (VTA), the nucleus accumbens (NAc) and the pre-frontal cortex (PFC) [22].

On the other hand, Alzheimer’s disease (AD) is a progressive neurodegenerative pathology with severe economic and social impact [23,24]. Nowadays, drug research and development are based on the cholinergic hypothesis which proposes that the selective loss of cholinergic neurons results in a deficit of ACh in specific regions of the brain (cerebral cortex and hippocampus) that mediate learning and memory functions. Based on this hypothesis, cholinergic augmentation will improve cognition in AD [25,26]. Currently available treatment for patients suffering from AD involves AChE inhibitors such as rivastigmine, donepezil, and galantamine, which avoid the hydrolysis of Ach, thereby increasing its concentration [27,28]. However, clinical efficiency is limited, as available AChE inhibitors can only ameliorate AD symptoms, and thus the search for novel compounds remains an emerging demand for the treatment of this pathology.

AD is associated with major serotonergic alterations due to involvement of the raphe nucleus and related projections. Additionally, both soluble and insoluble β-Amyloid (Aβ) species are associated with impaired synaptic plasticity and dysfunctional neurotransmission in serotonergic neurons [29,30] Furthermore, reductions in serotonin (5-HT, Figure 2) and its metabolite levels have been reported in brain tissue and cerebrospinal fluid in AD [31]. 

Studies carried out by Cirrito and Sheline [30,32] have demonstrated that activation of serotonergic neurotransmission may be beneficial in AD. The authors showed that acute administration of the selective 5-HT reuptake inhibitor (SSRI) citalopram in mice, at doses roughly equated to those prescribed for human patients with depression, reduced production of toxic Aβ proteins in the brain. Consistent with this finding, they further demonstrated that 5-HT infusion into the hippocampus of APP/PS1 mice also reduced the Aβ peptide in the brain. Clinical studies further support this idea, as Aβ imaging via positron emission tomography revealed lower cortical amyloid levels in study participants who had taken SSRIs within the past five years versus those who had not been treated with SSRIs.

Parkinson’s disease (PD) is a chronic, progressive neurodegenerative disease characterized by both motor and nonmotor features. The motor symptoms of PD are attributed to the loss of striatal dopaminergic neurons, although the presence of nonmotor symptoms supports neuronal loss in nondopaminergic areas as well [33]. Depression in PD patients is also attributed to serotonergic dysfunction, and SSRIs are able to amend depressive symptoms of PD [34]. Furthermore, several pieces of evidence confirm the neurodegeneration in striatal and extra-striatal 5-HT pathways suggesting that serotonergic loss plays an important role in the pathophysiology of PD [35,36]. Additionally, both nicotinic and muscarinic AChRs have been detected in the neostriatum (caudate nucleus and putamen), with the nAChRs activating the nigrostriatal dopamine (DA, Figure 2) release and the mAChRs causing inhibition of dopaminergic neurons [37]. 

All the aforementioned data are consistent with the well-established notion that monoaminergic and cholinergic neurotransmitter systems exhibit, in the central nervous system (CNS), a wide range of functional interactions and mutual regulations [38]. They also underline the complementary role that nAChRs and monoamines such as 5-HT and DA play in the modulation of several brain functions [39], as well as in the physiopathology of a variety of diseases [40]. 

In this context, it seems attractive to search for/formulate ligands that act through simultaneous interactions with SERT/DAT and nAChRs [41,42].

Therefore, based on the structural combination of nicotinic ligands and antidepressants, it seems attractive to search for and design compounds that act as multi-target therapeutic ligands in α4β2 nAChRs, DAT and/or SERT for the treatment of anxiety and depressive disorders [43,44].

Based on these precedents, in the present work, we report the synthesis and biochemical evaluation of a series of nicotine derivatives, where the *N*-methyl-pyrrolidine moiety of the parent drug was connected to aromatic and alkyl aromatic esters using a polymethylenic chain of variable length (n =1, 2). The rationale behind this design proposal is supported by pharmacological studies of our research group, in which a series of (S)-(1-methylpyrrolidin-2-yl)methyl benzoate derivatives exhibited antagonist activity on the α4β2 nAChRs [4]. Thus, to extend such study, we decided to synthesize novel molecules, which by the inclusion of aromatic moieties, might behave as multiligands, interacting simultaneously with the α4β2 nAChRs, DAT and/or SERT. Additionally, molecular docking was performed at the different studied targets to rationalize the affinity shown by some of our compounds.

## 2. Results and Discussion

### 2.1. Chemistry

The arylpyrrolidine ester derivatives, compounds (**1**–**21**) (Figure 3) were synthesized from the following commercially available compounds: benzoic acid and 2-phenylacetic acid derivatives, 2,2-diphenylacetyl chloride, 2-Naphthoyl chloride, (***S***)-(-)-(1-methyl-2-pyrrolidinyl)methanol, and (±)-2-(1-methyl-2-pyrrolidinyl)ethanol.

Figure 3 summarizes all the synthesized compounds, which were fully characterized spectroscopically by FT-IR, ^1^H-NMR, ^13^C-NMR and HRMS.

The substituted benzoic acids (A) were first reacted with thionyl chloride to give the corresponding benzoyl halides (B), which were subsequently reacted with (***S***)-(-)-(1-methyl-2-pyrrolidinyl)methanol and (±)-2-(1-methyl-2-pyrrolidinyl)ethanol as racemic mixture to give the corresponding final esters (**1**–**12**) in 61–81% yields (Appendix A).

The same reaction was carried out with the 2-phenylacetic acid derivatives (C) to provide the corresponding 2-arylacetylchlorides derivatives (D) (Appendix A), which were reacted with the commercially available (***S***)-(-)-(1-methyl-2-pyrrolidinyl)methanol and (±)-2-(1-methyl-2-pyrrolidinyl)ethanol as racemic mixture, to furnish the expected esters (**13**–**17**) in 20–35% yields (Appendix A).

Analogously the 2-Naphthoyl chloride, reacted with (***S***)-(-)-(1-methyl-2-pyrrolidinyl) methanol and (±)-2-(1-methyl-2-pyrrolidinyl)ethanol to give the corresponding final esters (**18**–**19**) in 61–65% yields (Appendix A).

Finally, the 2,2-diphenylacetyl chloride was reacted with (***S***)-(-)-(1-methyl-2-pyrrolidinyl)methanol and (±)-2-(1-methyl-2-pyrrolidinyl)ethanol as racemic mixture, to obtain the corresponding final esters (**20**–**21**) in 31–43% yields (Appendix A).

### 2.2. Biological Evaluation: Binding Affinities on h-DAT, h-SERT and α4β2 nAChR

To evaluate the affinity (K_i_) of our compounds, competitive binding assays in α4β2 nAChR using [^3^H]-cytisine as radioligand ([^3^H]-Cyt) was measured. Binding experiments were carried out on whole rat brain synaptosomes. To evaluate the affinity on the monoamine transporters SERT and DAT, binding affinity (K_i_) was determined using [^3^H]-WIN 35,428 ([^3^H]-WIN) for h-DAT and [^3^H]-paroxetine ([^3^H]-parox) for h-SERT as specific radioligands for the corresponding monoamine transporters respectively. Competitive binding studies were carried out on homogenized membranes prepared from the human clonal cell line HEK293 for h-SERT (Perkin Elmer) and CHO-K1 for h-DAT (Perkin Elmer). The estimated K_i_ values indicate that some compounds display competitive binding affinity for h-DAT and α4β2 nAChR. All compounds were ineffective to displace [^3^H]-paroxetine from h-SERT. However, an unexpected result was found with compound **20**, which produced a strong increase of affinity for [^3^H]-paroxetine. K_i_ values for α4β2 nACHR, h-DAT and h-SERT are shown in Table 1.

Binding experiments for h-SERT show that all compounds are unable to displace [^3^H]-paroxetine from its binding site, but unexpectedly, an increase of radioactivity of over 100% was found for some values of paroxetine binding. These increases in the binding of the radioligand did not exceed 40% in most of cases. However, compound **20** elicited an increase of 91% of the total radioactivity binding, indicating a possible allosteric interaction of this compound, which could be responsible for the increase of total [^3^H]-paroxetine bonded at the h-SERT binding site (Figure 4). 

Compounds **2** and **4** displayed lower affinity in α4β2 nAChR (K_i_ = 0.094 ± 0.002 µM and K_i_ = 0.132 ± 0.038 µM) as compared with the unsubstituted analog **1,** but induced an increase in the percentage binding of [^3^H]-paroxetine at h-SERT (123.0 ± 1.2% and 137.0 ± 1.2%), suggesting that the presence of aromatic substituents 3-F and 3-Br would disfavor the interaction in the agonist binding site at α4β2 nAChRs, but would favor the interactions at level of a putative allosteric binding site of h-SERT. Compound **3** with a K_i_ of 0.009 ± 0.001 µM on the α4β2 nAChRs (Figure 5) lacks affinity at h-SERT and h-DAT. In comparison, compounds **5** and **6** (not included in Table 1) did not display affinity for any of the binding sites under consideration, suggesting that methyl substituents at C-2 or C-4 of the aromatic ring lead to compounds with a total lack of molecular interactions at these targets.

Compound **7**, in comparison with compound **1**, showed a low affinity for α4β2 nAChR (K_i_ = 1.788 ± 0.378 µM) suggesting that when the (***S***)-(1-methyl-2-pyrrolidinyl)methanol moiety was changed for (±)-(1-methyl-2-pyrrolidinyl)ethanol, the interaction with the agonist binding site α4β2 nAChR is remarkably disfavored (Figure 5). In addition, no affinity for h-SERT and h-DAT was observed for this derivative. Compound **9** gave a comparative low affinity for α4β2 nAChR (K_i_ of 3.461 ± 0.360 µM, Figure 5), whereas compound **10** resulted in be one of the best of this series, showing a high affinity for α4β2 nAChR (K_i_ of 0.042 ± 0.004 µM) and eliciting also an increase in the percentage of union of [^3^H]-paroxetine at h-SERT (119.3 ± 2.7%). Compounds **11** and **12** (not included in Table 1) similarly to compounds **5** and **6**, did not display affinity for any of the binding sites under consideration, suggesting again that methyl substituents at C-2 or C-4 of the aromatic ring lead to a total lack of molecular interactions.

The unsubstituted compound **16** showed a relatively low affinity for h-DAT (K_i_ = 22.690 ± 7.099 µM), whereas compound **17**, with a 3,4-dichloro substitution pattern, was approximately 7-fold more potent at h-DAT with a K_i_ of 3.317 ± 0.923 µM. This indicates that the halogenation of compound **16** at C-3 and C-4 positions, favors the ligand interactions with the central binding site of h-DAT. Compound **18** exhibited affinities for α4β2 nAChR (K_i_ = 0.120 ± 0.037 μM) and for h-DAT (K_i_ = 99.330 ± 1.411 μM), and also produced a small increase of the percent of [^3^H]-paroxetine binding at h-SERT (117.0 ± 2.3%). Compound **19** also produced an increase in the percent binding of [^3^H]-paroxetine at h-SERT (126.3 ± 1.9%) and showed h-DAT affinity (K_i_ = 44.240 ± 8.120 μM). 

Compounds **18** and **19** were less potent than compounds **16** and **17** at h-DAT, indicating that the incorporation of a naphthyl moiety (compounds **18** and **19**) decreases the affinity for this target (Figure 6). On the other hand, the ethylene linker connecting the pyrrolidine moiety and the aryl ester in **16** and **17** improved the affinity as compared with compounds **13**, **14** and **15** (not included in Table 1) which contained a methylene linker.

On the other hand, compound **20** was the only compound of the series which displayed a triple affinity for the studied targets. Thus, this compound produced a marked increase in the percent of [^3^H]-paroxetine binding at h-SERT (191.5 ± 0.9%) (Figure 4), showed high affinity at α4β2 nAChR (K_i_ = 0.023 ± 0.006 μM) (Figure 5) and a moderate affinity for h-DAT (K_i_ = 1.208 ± 0.230 μM) (Figure 6). Compound **21** was the most potent of the series at h-DAT, exhibiting an affinity at this target which was higher than that of bupropion (K_i_ = 0.075 ± 0.009 μM vs. K_i_ = 0.370 μM respectively). Interestingly, compound **21** exhibited a similar affinity at α4β2 nAChR (K_i_ = 0.113 ± 0.037 μM), which was also higher than that of bupropion at this receptor (K_i_ = 10 μM). Thus, **21** can be considered as the compound with the best dual affinity of the series. These biphenyl analogs (**20** and **21**), were the most potent compounds in h-DAT binding experiments. However, compound **21** having both an ethylenic bridge and a diphenyl ester moiety, contains the two pharmacophore portions for optimal binding affinity on h-DAT (Figure 6).

In an related work, Zhang et al. [12] described a series of novel ligands containing cyclopropane- or isoxazole- side chains onto the 5-position of 1-(pyridin-3-yl)-1,4-diazepine or 2-(pyridin-3-yl)-2,5-diazabicyclo [2.2.1]heptane which afforded compounds with selective partial agonist propertiers for α4β2* nAChR (K_i_ of 0.5-51.4 nM), displaying also excellent selectivity over the α3β4 subtype. They also found that the most potent compounds, structurally exhibited a pyridine ring connected to a diazepine squeleton, where substituents at the 5-position of the pyridine core were found to be beneficial for the selectivity for α4β2 over α7 and ganglionic nAChRs. In connection with the Zhang approaches, we carried out the synthesis of molecules bearing a pyridinyl group connected to an ester function. However, in our case these modifications lead to unfavorable interactions at the agonist binding site for both α4β2 and α7 nAChRs [4]. The change of the azaheterocycle ring by a functionalized phenyl group afforded the results already described.

In recent studies, we have replaced the ester function for an ether function [43], obtaining molecules with agonist activity at α4β2* nAChR which in theory promote dopamine release and would not be useful in nicotine addiction. 

### 2.3. Docking Studies

To gain a better insight into the binding mode and possible interactions of the compounds into the target binding sites, we carried out docking studies in a homology model of the agonist site of the α4β2 nAChR, the central site of the DAT, and the best characterized allosteric site (the S2 site) of SERT, using AutoDock 4.0.

Binding studies with α4β2 nAChRs indicated that only 10 compounds (**1**–**21**) showed affinity at the ACh binding site of α4β2 nAChRs, i.e the orthosteric site (values ranging from low micro to submicromolar concentrations). Docking studies are in line with these results, mainly due to the observed π–cation interaction with Trp 182 residue, retained in all the nicotine-derived synthetic compounds and bupropion. By comparison with bupropion, the higher affinities of some of our compounds, could be explained by both the presence of hydrogen bond and hydrophobic interactions, which would take place between the aromatic ring with Tyr 223 or Phe 119 residues (compounds **2**, **4**, **18** and **20**) and between the quaternary nitrogen of the *N*-methylpyrrolidine with Tyr 126 residue (compound **1**). On the other hand, compound **3** exhibited an additional π–cation interaction between the quaternary nitrogen of the *N*-methylpyrrolidine with the ester group of the Tyr 230 residue. In the docking studies, the high-affinity obtained for compound **21** (Figure 7B), could be explained due to the presence of π–π interactions among the Phe 119 residue with both phenyl groups of compound **21**, increasing the energy of the ligand-receptor complex (Figure 7B), these interactions are absent in bupropion as was already mentioned [45].

The binding studies carried out in SERT that all compounds were unable to displace [^3^H]-paroxetine from its binding site, but unexpectedly compound **20** exhibited an increase of the radioactivity, indicating a possible allosteric interaction in SERT. Docking studies are in line with this result, mainly due to the presence of an ionic bond and π-stacking interactions between the quaternary nitrogen with Glu 494 and Phe 335 residues, respectively. These interactions would take place in both compound **20** and the S-citalopram co-crystallized with SERT (Figure 8B). Additionally, compound **20** exhibited the highest affinity in the allosteric h-SERT binding site, probably due to a new π-stacking interaction between one of the phenyl group of **20** with Phe 556 and a π–cation interaction between the phenyl group of **20** and Arg 104 residue, which were not observed in the S-citalopram co-crystal (Figure 8B).

Binding studies in h-DAT indicate that only 6 compounds (**1**–**21**) showed affinity values at the central binding site of h-DAT. The results from the docking studies correlated with experimental evidence, primarily due to the presence of a π–π interaction between the aromatic group of compounds **18, 19** and **20** and the Tyr 548 residue, and a π–cation interaction among the phenyl groups of compounds **16** and **19** with the quaternary amine of Ala 550, both interactions retained for neo-nicotinic compounds and bupropion. Compound **17** showed a double hydrogen bond interaction, among Tyr 470 residue with the quaternary nitrogen of the *N*-methylpyrrolidine and the ester function of **17**, with both interactions absent in bupropion. Compound **21** showed the highest affinity in relation to bupropion. This experimental value can be supported by the docking results, on the basis of the following interactions observed: a π–π type interaction between both phenyl groups and Tyr 470 residue, a π–π interaction between one of the phenyl group and Tyr 551 residue, and two π–cation interactions between the phenyl groups with Ser 227 and Tyr 548 residues. On the other hand, compound **21** showed a hydrogen bond interaction between the quaternary amine group of *N*-methylpyrrolidine and Tyr 548 residue, and three hydrogen bond interactions between the ester function of **21** with Ala 550 and Tyr 551 residues, contributing in consequence to the thermodynamic stability of the system (Figure 9B).

## 3. Materials and Methods

### 3.1. Binding Protocol of [^3^H]-Cytisine on nAChR Synaptosomes

Preparation of the synaptosome fraction: the rats were sacrificed by decapitation, and their brain removed without dissecting. Brain tissue was homogenized in 40 volumes using a buffer of 0.32 M sucrose at pH 7.4, containing 1 mM EDTA, 0.1 mM PMSF, and 0.01% of cold and freshly prepared NaN_3,_ through a glass homogenizer and teflon stem [46]. The homogenate was distributed in Eppendorf tubes and centrifuged at 20,000 g for 30 min at 4 °C to produce a synaptosome pellet (fraction P1) which is used for the binding assay. This fraction P1 was subsequently stored at −80 °C. 

Binding assay of [^3^H]-cytisine in α4β2 nAChR cells*:* the rat complete brain P1 fraction (occupying 2 mg of wet tissue for each assay) was incubated in 2500 μL of 20 mM HEPES buffer solution at pH 7.4 containing 118 mm NaCl, 4.8 mm KCl, 2.5 mm CaCl_2_, 200 mm Tris, 0.1 mm PMSF and 0.01% NaN_3_ with or without the different drugs under study at different concentrations, in the presence of 1 nM of [^3^H]-cytisine (specific activity 35.8 Ci/mmol, Code NET1054025UC, Perkin-Elmer, Santiago, Chile) in a final volume of 250 μL. Non-specific binding was determined using 2.5 mM of cytisine. Incubation was performed for 30 min at room temperature and then for 60 min at 4 °C. Incubation was stopped by rapid filtration in Whatman GF/C filter preabsorbed in 0.5% polyethylenimine, which were washed with cold and fresh working buffer 7 times, then dried, and adding scintillation liquid, left for the whole night in Eppendorf tubes in order to measure later the radioactivity by liquid scintillation spectrometry (MicroBeta 2450 microplate counter, PerkinElmer, Santiago, Chile). The data were plotted by non-linear regression variable inhibitor-response dose (Prism 5.01, GraphPad, San Diego, CA, USA) to obtain the IC_50_ and K_i_ values of the tested compounds using the Cheng-Prusoff equation.

### 3.2. Protocol Binding of [^3^H]-Paroxetine on h-SERT Cells

The cellular background HEK293 containing 400 μL of h-SERT (Code RBHSTM400UA, Perkin-Elmer, Santiago, Chile) was diluted in 12 Eppendorf tubes, containing a storage buffer solution of Tris-HCl 50 mM (pH 7.4), EDTA 0.5 mM, MgCl_2_ 10 mM and 10% sucrose, obtaining a final volume between 260 and 340 μL, which was finally stored at −80 °C [46].

Each Eppendorf tube was incubated with 50 mM Tris HCl buffer (pH 7.4), 120 mM NaCl, 5 mM KCl and the drugs under study using increasing concentrations, in presence of 2 nM of [^3^H]-paroxetine (specific activity 23.1 Ci/mmol, Code NET86925UC, Perkin-Elmer, Santiago, Chile) with a final volume of 250 μL [46]. Non-specific binding was determined using 25 mM fluoxetine. After 30 min at 27 °C, the incubation was stopped by rapid filtration on a Whatman GF/C filter preabsorbed in 0.5% polyethylenimine (PEI), washed with cold working buffer solution, 3 × 3mL, filtered, and scintillation liquid was added. The radioactivity was measured by liquid scintillation spectrometry (MicroBeta 2450 microplate counter, PerkinElmer, Santiago, Chile). The data were represented in a bar graph using 25 μM concentration for both, the compound under study and fluoxetine as the non-specific ligand. Each bar represents the mean ± S.E.M obtained in the experiments carried out in triplicates.

### 3.3. Protocol Binding of [^3^H]-WIN 35,428 on h-DAT Cells

The cellular background CHO-K1 with 400 μL of h-DAT (Code RBHDATM400UA, Perkin-Elmer, Santiago, Chile) was diluted in 12 Eppendorf tubes, with a storage buffer solution of Tris-HCl 50 mM (pH 7.4), EDTA 0.5 mM, MgCl_2_ 10 mM and 10% sucrose, obtaining a final volume of 260–340 μL stored at −80 °C [46]. Each Eppendorf tube was incubated in 50 mM Tris-HCl buffer solution (pH 7.4), 100 mM NaCl, the drugs under study were using increasing concentrations, with 1 nM of [^3^H]-WIN 35,428 (specific activity 82.6 Ci/mmol, Code NET1033025UC, Perkin-Elmer, Santiago, Chile) with a final volume of 250 μL. Non-specific binding was determined using 10 mM methylphenidate.

After 120 min at 4 °C, the incubation was stopped by rapid filtration on a Whatman GF/C filter preabsorbed in 0.5% polyethylenimine (PEI), washed with cold working buffer solution, 3 x 3mL, filtered, and scintillation liquid was added. The radioactivity was measured by liquid scintillation spectrometry (MicroBeta 2450 microplate counter, PerkinElmer, Santiago, Chile). The data were plotted by non-linear regression variable inhibitor-response dose (Prism 5.01, GraphPad, San Diego, CA, USA) to estimate the IC_50_ and K_i_ values for the tested compounds using the Cheng-Prusoff equation.

### 3.4. Docking Analysis 

Molecular docking of the compounds in the homology model of the α4β2, central binding site of h-DAT and allosteric binding site of h-SERT was performed, using the Lamarckian genetic algorithm search method with the AutoDock v4.0 software, San Diego, CA, USA [47]. The homology models of the proteins were obtained from the Protein Data Bank (PDB: 5I73, 1UW6 and 4M48 respectively). The receptors were kept rigid, while full flexibility was allowed for the ligands to translate/rotate. Polar hydrogens were added to the receptors and Kollman-united atom partial charges along with atomic solvation parameters were assigned to the individual protein atoms. The three-dimensional structures of each ligand were generated using the SPARTAN’08 program and energy minimized. For each ligand, a rigid root and rotatable bonds were assigned automatically. The non-polar hydrogens were removed and the partial charges from these were added to the carbons (Gasteiger charges). The atom type for aromatic carbons was reassigned in order to use the AutoDock v4.0 aromatic carbon grid map. Docking was carried out using 60 × 60 × 60 grid points with a default spacing of 0.375 Å [48]. For the nAChR, the grid was positioned to include the full ligand binding pocket in the central part of the α4/β2 subunit interfaces to allow extensive sampling around residue α4W182. For the allosteric binding site of the h-SERT a radio lattice of 14 Å was defined around co-crystal (*S*-Citalopram) and was the only established restriction. The hydrogen bond generation between the carboxylate group of E494 residue was conserved for the majority of the known complexes studied between h-SERT and S-citalopram [49]. For the binding of h-DAT, the grid was positioned to include the full ligand binding pocket in the central part of the protein to allow extensive sampling around the residue Y548**.** Within this grid, the Lamarckian genetic search algorithm was used with a population size of 150 individuals, calculated using 200 different runs (i.e., 200 dockings). Each run had two stop criteria, a maximum of 1.5 × 10^6^ energy evaluations or a maximum of 50,000 generations, starting from a random position and conformation; default parameters were used for the Lamarckian genetic algorithm search. The optimal configuration and the resulting ligand–receptor complexes were further processed using the PYMOL software, New York, NY, USA [48].

## 4. Synthetic Procedures

Melting points are uncorrected and were determined with a Reichert Galen III hot plate microscope. ^1^H-NMR and ^13^C-RMN spectra were recorded using Bruker AMX 400 spectrometers at 400 MHz. The ^1^H-NMR and ^13^C-NMR chemical shifts were referenced with respect to residual solvent peaks (δ TMS = 0) and all quoted coupling constants J are J_HH_ between protons. The IR spectra were recorded on a FT-IR Nicolet iS50 Thermo scientific and wavenumbers are reported in cm^−1^. High resolution mass spectra were recorded on a DSA–TOF AxION 2 TOF MS (Perkin Elmer, Shelton, CT, USA), under positive mode, the molecular ion [M+1] is observed. 

Reactions and product mixtures were routinely monitored by thin-layer chromatography (TLC) on silica gel pre-coated F_254_ Merck plates and the compounds obtained were purified by column chromatography using a CH_2_Cl_2_/CH_3_OH (9:1) mixture as the mobile phase. Reagents and solvents utilized were commercially available and used without further purification.

### 4.1. General Procedures for The synthesis of Benzoyl Chloride and 2-Phenylacetyl Chloride Derivatives *(**1**–**12**)*

Benzoic acid derivatives (**1**–**6**) and 2-phenylacetic acid derivatives (**7**–**12**) (6–10 mmol, 0.5 g) were reacted with thionyl chloride (68.9 mmol, 5 mL) under nitrogen atmosphere for 24 h to afford the corresponding benzoyl chloride or 2-phenylacetyl chloride derivatives. The mixture was maintained for 15 min. at room temperature and then concentrated under vacuum to give the corresponding crude aroyl chloride derivatives, which were immediately used in the next reaction.

### 4.2. General Procedures for the Synthesis of (S)-(1-methyl-2-pyrrolidinyl)methyl Benzoate Derivatives *(**1–6**)* and (S)-(1-methyl-2-pyrrolidinyl)methyl 2-naphtoate *(**18**)*

Benzoyl chloride derivatives (4.2 mmol, 0.5 mL) or 2-Naphthoyl chloride (4.2 mmol, 0.80 g) were dissolved in dry diethyl ether (50 mL) and stirred at room temperature. One equivalent of (***S***)-(-)-(1-methyl-2-pyrrolidinyl)methanol (2.2–4.2 mmol, 0.2–0.5 mL) in dry diethyl ether (30 mL) was added drop by drop at room temperature. The reaction mixture was kept at room temperature with stirring for 24 h. Then, the mixture was concentrated under vacuum to give a crude which was re-dissolved in water adjusted to pH 8.0 and extracted with CH_2_Cl_2_ (3 × 30 mL). Finally, the hydrochloride salts were obtained from an isopropanol solution 7.4% HCl (*p*/*v*).

### 4.3. General Procedures for the Synthesis of (S)-(1-methyl-2-pyrrolidinyl)methyl 2-phenylacetate Derivatives *(**13**–**15**)* and (S)-(1-methyl-2-pyrrolidinyl)methyl 2,2-diphenylacetate *(**20**)*

2-Phenylacetyl chloride derivatives and 2,2-diphenylacetyl chloride (2.5–4.2 mmol, 0.5 gr) were dissolved in dry diethyl ether (50 mL) and stirred at room temperature. One equivalent of (***S***)-(-)-(1-methyl-2-pyrrolidinyl)methanol (2.2–4.2 mmol, 0.2–0.5 mL) in dry diethyl ether (30 mL) was slowly added. The reaction mixture was kept at room temperature with stirring for 24 h. Then, the mixture was concentrated under vacuum to give a crude which was re-dissolved in water adjusted to pH 8.0 and extracted with CH_2_Cl_2_ (3 × 30 mL). Finally, the hydrochloride salts were obtained from an isopropanol solution 7.4% HCl (*p*/*v*).

### 4.4. General Procedures for the Synthesis of (±)-2-(1-methyl-2-pyrrolidinyl)ethyl Benzoate Derivates *(**7**–**12**)* and (±)-2-(1-methyl-2-pyrrolidinyl)ethyl 2-naphtoate *(**19**)*

Benzoyl chloride derivatives (3.7 mmol, 0.4–0.5 mL) and 2-naphthoyl chloride (3.7 mmol, 0.7 g) were dissolved in dry diethyl ether (50 mL) and stirred at room temperature. One equivalent of (±)-2-(1-methyl-2-pyrrolidinyl)ethanol (3.7 mmol, 0.5 mL) in diethyl ether (30 mL) was added drop by drop. The reaction mixture was kept at room temperature and stirred for 24 h. Then, the mixture was concentrated under vacuum to give a crude which was re-dissolved in water, adjusted to pH 8.0 and extracted with CH_2_Cl_2_ (3 × 30 mL). The hydrochloride salts were obtained from an isopropanol solution 7.4% HCl (*p*/*v*).

### 4.5. General Procedures for the Synthesis of (±)-2-(1-methyl-2-pyrrolidinyl)ethyl 2-phenylacetate Derivatives *(**16**–**17**)* and (±)-2-(1-methyl-2-pyrrolidinyl)ethyl 2,2-diphenylacetate *(**21**)*

2-Phenylacetyl chloride derivatives and 2,2-diphenylacetyl (2.2–4.1 mmol, 0.3–0.5 mL) was dissolved in dry diethyl ether (50 mL) and stirred at room temperature. An equivalent of (±)-2-(1-methylpyrrolidin-2-yl)ethanol (2–4 mmol, 0.3–0.5 mL) in diethyl ether (30 mL) was added dropwise. The reaction mixture was kept at room temperature and stirred for 24 h. Then, the mixture was concentrated under vacuum to give a crude which was re-dissolved in water adjusted to pH 8.0 and extracted with CH_2_Cl_2_ (3 × 30 mL). The hydrochloride salts were obtained from an isopropanol solution 7.4% HCl (*p*/*v*).

### 4.6. (S)-(1-Methyl-2-pyrrolidinyl)methyl Benzoate *(**1**)*

Prepared as described in general procedure (Section 4.2). (***S***)-(-)-(1-methyl-2-pyrrolidinyl)methanol (4.2 mmol, 0.42 mL) was added to a solution of benzoyl chloride (4.2 mmol, 0.49 mL); Yield 718 mg (78%); m.p. 165.4–166.0 °C; IR (cm^−1^) 2908 (stretch. C-H aliph.), 1754 (stretching COO), 1280 (stretching C-O), 1115 (stretching C-N), 716 (bending C-H Arom.); ^1^H-NMR (400 MHz, D_2_O) δ 8.08 (d, *J* = 7.8 Hz, 2H), 7.75 (t, *J* = 7.5 Hz, 1H), 7.59 (t, *J* = 7.7 Hz, 2H), 4.76 (d, *J* = 2.9 Hz, 1H), 4.58 (dd, *J* = 13.1, *J* = 6.7 Hz, 1H), 3.95 (m, 1H), 3.78 (m, 1H), 3.29 (m, 1H), 3.08 (s, 3H), 2.45 (m, 1H), 2.26 (m, 1H), 2.12 (m, 2H); ^13^C-RMN (101 MHz, D_2_O) δ 165.3, 133.6, 129.4 (2C), 129.1, 128.7 (2C), 66.0, 62.5, 56.1, 39.5, 26.6, 21.7; HRMS *m/z* calcd. for C_13_H_17_NO_2_ (M^+^), 220.1338; found, 220.1347.

### 4.7. (S)-(1-Methyl-2-pyrrolidinyl)methyl 3-fluorobenzoate *(**2**)*

Prepared as described in general procedure (Section 4.2). (***S***)-(-)-(1-methyl-2-pyrrolidinyl)methanol (4.2 mmol, 0.5 mL) was added to a solution of 3-fluorobenzoyl chloride (4.2 mmol, 0.5 mL); Yield 807 mg (81%); m.p. 110–111.9 °C; IR (cm^−1^) 2965 (stretch. C-H aliph.), 1724 (stretching COO), 1292 (stretching C-O), 1071 (stretching C-N), 757 (bending C-H Arom.); ^1^H-NMR (400 MHz, DMSO-*d*_6_) δ: 7.90 (d, *J* = 7.5 Hz, 1H), 7.79 (d, *J* = 7.5 Hz, 1H), 7.56 (m, 1H), 7.48 (t, *J* = 8.5 Hz, 1H), 4.63 (m, 2H), 3.81 (m, 1H), 3.56 (m, 1H), 3.13 (m, 1H), 2.87 (s, 3H), 2.25 (m, 1H), 2.00 (m, 2H), 1.87 (m, 1H); ^13^C-RMN (101 MHz, DMSO-*d*_6_) δ: 167.9, 165.1-162.6, 132.4 (2C), 127.2, 122.7, 117.8, 68.9, 64.2, 58.8, 42.2, 27.7, 23.6; HRMS *m/z* calcd. for C_13_H_16_FNO_2_ (M^+^), 238.1248; found, 238.1266.

### 4.8. (S)-(1-Methyl-2-pyrrolidinyl)methyl 3-chlorobenzoate *(**3**)*

Prepared as described above in general procedure (Section 4.2). (***S***)-(-)-(1-methyl-2-pyrrolidinyl)methanol (4.2 mmol, 0.5 mL) was added to a solution of 3-chlorobenzoyl chloride (4.2 mmol, 0.54 mL); Yield 650 mg (61%); m.p. 120.5–121.9 °C; IR (cm^−1^) 2945 (stretch. C-H aliph.), 1737 (stretching COO), 1260 (stretching C-O), 1143 (stretching C-N), 748 (bending C-H Arom.); ^1^H-NMR (400 MHz, DMSO-*d*_6_) δ: 8.02 (d, *J* = 13.3 Hz, 2H), 7.77 (d, *J* = 8.0 Hz, 1H), 7.59 (t, *J* = 7.8 Hz, 1H), 4.63 (m, 2H) 3.80 (m, 1H), 3.58 (m, 1H), 3.09 (d, *J* = 9.0 Hz, 1H), 2.83 (s, 3H), 2.24 (m, 1H), 2.05 (m, 2H), 1.92 (m, 1H); ^13^C-NMR (101 MHz, DMSO-*d*_6_) δ: 166.8, 135.3, 135.2 (2C), 131.9, 130.5, 129.4, 70.8, 59.7, 57.8, 41.3, 26.98, 22.95; HRMS *m/z* calcd. for C_13_H_16_ClNO_2_ (M^+^), 254.0948; found, 254.0967.

### 4.9. (S)-(1-Methyl-2-pyrrolidinyl)methyl 3-bromobenzoate *(**4**)*

Prepared as described above in general procedure (Section 4.2). (***S***)-(-)-(1-methyl-2-pyrrolidinyl)methanol (4.2 mmol, 0.5 mL) was added to a solution of 3-bromobenzoyl chloride (4.2 mmol, 0.55 mL); Yield 914 mg (73%); m.p. 99–101,4 °C; IR (cm^−1^) 3069 (stretch. C-H aliph.), 1744 (stretching COO), 1208 (stretching C-O), 1074 (stretching C-N), 728 (bending C-H Arom.); ^1^H-NMR (400 MHz, DMSO-*d*_6_) δ: 8.03 (s, 1H), 7.93 (d, *J* = 7.7 Hz, 1H), 7.82 (d, *J* = 7.9 Hz, 1H), 7.47 (t, *J* = 7.8 Hz 1H), 4.60 (dt, *J* = 12.7, 9.6 Hz, 2H), 3.82 (t, *J* = 6.88, 6.88 Hz, 1H), 3.58 (m, 1H), 3.15 (m, 1H), 2.90 (s, 3H), 2.23 (m, 1H), 1.94 (m, 3H); ^13^C-NMR (101 MHz, D_2_O) δ: 165.2, 135.9, 131.2, 129.5, 129.4, 127.3, 121.0, 66.4, 61.5, 56.2, 39.6, 25.1, 20.9; HRMS *m/z* calcd. for C_13_H_16_BrNO_2_ (M^+^), 298.0438; found, 298.0462.

### 4.10. (S)-(1-Methyl-2-pyrrolidinyl)methyl 2-methylbenzoate *(**5**)*

Prepared as described above in general procedure (Section 4.2). (***S***)-(-)-(1-methyl-2-pyrrolidinyl)methanol (4.2 mmol, 0.5 mL) was added to a solution of 2-methylbenzoyl chloride (4.2 mmol, 0.55 mL); Yield 725 mg (74%); m.p. 130.5–131.6 °C; IR (cm^−1^) 2948 (stretch. C-H aliph.), 1725 (stretching COO), 1280 (stretching C-O), 1126 (stretching C-N), 757 (bending C-H Arom.); ^1^H-NMR (400 MHz, DMSO-*d*_6_) δ: 7.96 (d, *J* = 7.7 Hz, 1H), 7.51 (t, *J* =7.34 Hz, 1H), 7.34 (m, 2H), 4.59 (m, 2H), 3.76 (m, 1H), 3.53 (m, 1H), 3.09 (m, 1H), 2.87 (s, 3H), 2.54 (s, 3H) 2.26 (m, 1H), 1.99 (m, 2H), 1.87 (m, 1H); ^13^C-NMR (101 MHz, DMSO-*d*_6_) δ: 169.3, 141.8, 134.7, 133.3, 131.8, 127.8, 127.6, 68.6, 63.7, 58.5, 42.1, 27.7, 23.5, 22.4; HRMS *m/z* calcd. for C_14_H_19_NO_2_ (M^+^), 234.1498; found, 234.1494.

### 4.11. (S)-(1-Methyl-2-pyrrolidinyl)methyl 4-methylbenzoate *(**6**)*

Prepared as described above in general procedure (Section 4.2). (***S***)-(-)-(1-methyl-2-pyrrolidinyl)methanol (4.2 mmol, 0.5 mL) was added to a solution of 4-methylbenzoyl chloride (4.2 mmol, 0.55 mL); Yield 774 mg (79%); m.p. 158.1–159.4 °C; IR (cm^−1^) 2940 (stretch. C-H aliph.), 1719 (stretching COO), 1268 (stretching C-O), 1103 (stretching C-N), 752 (bending C-H Arom.); ^1^H-NMR (400 MHz, DMSO-*d*_6_) δ: 7.93 (d, *J* = 7.7 Hz, 2H), 7.35 (d, *J* = 7.8 Hz, 2H), 4.60 (m, 2H), 3.79 (m, 1H), 3.58 (m, 1H), 3.09 (m, 1H), 2.90 (s, 3H), 2.39 (s, 3H), 2.24 (m, 1H), 1.98 (m, 2H), 1.84 (m, 1H); ^13^C-NMR (101 MHz, DMSO-*d*_6_) δ: 168.7, 146.9, 131.0 (2C), 130.9 (2C), 126.9, 68.7, 63.6, 58.5, 41.9, 27.5, 23.4, 22.3; HRMS *m/z* calcd. for C_14_H_19_NO_2_ (M^+^), 234.1498; found, 234.1495.

### 4.12. (±)-2-(1-Methyl-2-pyrrolidinyl)ethyl benzoate *(**7**)*

Prepared as described above in general procedure (Section 4.4). (±)-2-(1-methyl-2-pyrrolidinyl)ethanol (3.7 mmol, 0.5 mL) was added to a solution of benzoyl chloride (3.7 mmol, 0.43 mL); Yield 641 mg (74.6%); m.p. 88.6–89.1 °C; IR (cm^−1^) 2953 (stretch. C-H aliph.), 1716 (stretching COO), 1280 (stretching C-O), 1115 (stretching C-N), 711 (bending C-H Arom.); ^1^H-NMR (400 MHz, D_2_O) δ 8.04 (d, *J* = 7.7 Hz, 2H), 7.72 (t, *J* = 7.5 Hz, 1H), 7.57 (t, *J* = 7.5 Hz, 2H), 4.47 (m, 2H), 3.72 (m, 1H), 3.52 (m, 1H), 3.18 (m, 1H), 2.98 (s, 3H), 2.46 (m, 3H), 2.13 (dt, *J* = 11.16, 6.06, 6.06 Hz, 1H), 1.87 (m, 2H); ^13^C-RMN (101 MHz, DMSO-*d*_6_) δ: 165.7, 133.5, 129.5, 129.3 (2C), 128.8 (2C), 65.4, 62.0, 57.6, 38.1, 29.4, 28.8, 21.1; HRMS *m/z* calcd. for C_14_H_19_NO_2_ (M^+^), 234.1498; found, 234.1493.

### 4.13. (±)-2-(1-Methyl-2-pyrrolidinyl)ethyl 3-fluorobenzoate *(**8**)*

Prepared as described above in general procedure (Section 4.4). (±)-2-(1-methyl-2-pyrrolidinyl)ethanol (3.7 mmol, 0.5 mL) was added to a solution of 3-bromobenzoyl chloride (3.7 mmol, 0.44 mL); Yield 636 mg (68.8%); m.p. 106.8–107.5 °C; IR (cm^−1^) 2965 (stretch. C-H aliph.), 1721 (stretching COO), 1280 (stretching C-O), 1082 (stretching C-N), 760 (bending C-H Arom.); ^1^H-NMR (400 MHz, D_2_O) δ 7.84 (d, *J* = 7.7 Hz, 1H), 7.72 (d, *J* = 9.7 Hz, 1H), 7.55 (dd, *J* = 13.9, *J* = 7.9 Hz, 1H), 7.45 (dt, *J* = 8.7, 1.7 Hz, 1H), 4.49 (m, 2H), 3.75 (m, 1H), 3.53 (m, 1H), 3.20 (dt, *J* = 11.5, 8.5 Hz, 1H), 2.99 (s, 3H), 2.51 (m, 2H), 2.13 (m, 3H), 1.91 (m, 1H); ^13^C-RMN (101 MHz, DMSO-*d*_6_) δ: 164.5, 163.2-160.8, 131.9, 131.1, 125.5, 120.5, 115.8, 65.2, 62.3, 54.7, 38.1, 29.2, 28.9, 20.9; HRMS *m/z* calcd. for C_14_H_18_FNO_2_ (M^+^), 252.1398; found, 252.1403.

### 4.14. (±)-2-(1-Methyl-2-pyrrolidinyl)ethyl 3-chlorobenzoate *(**9**)*

Prepared as described above in general procedure (Section 4.4). (±)-2-(1-methyl-2-pyrrolidinyl)ethanol (3.7 mmol, 0.5 mL) was added to a solution of 3-chlorobenzoyl chloride (3.7 mmol, 0.47 mL); Yield 680 mg (69.0%); m.p. 144.7–146.2 °C; IR (cm^−1^) 2916 (stretch. C-H aliph.), 1724 (stretching COO), 1268 (stretching C-O), 1074 (stretching C-N), 750 (bending C-H Arom.); ^1^H-NMR (400 MHz, D_2_O) δ 7.85 (d, *J* = 11.5 Hz, 2H), 7.64 (d, *J* = 8.0 Hz, 1H), 7.46 (t, *J* = 7.9 Hz, 1H), 4.47 (m, 2H), 3.74 (m, 1H), 3.54 (m, 1H), 3.21 (m, 1H), 2.99 (s, 3H), 2.47 (m, 2H), 2.14 (m, 3H), 1.89 (m, 1H); ^13^C-NMR (101 MHz, DMSO-*d*_6_) δ: 164.4, 133.5, 133.3, 131.6, 130.8, 128.7, 127.9, 65.2, 62.4, 54.7, 38.1, 29.2, 28.6, 21.0; HRMS *m/z* calcd. for C_14_H_18_ClNO_2_ (M^+^), 268.1108; found, 268.1114.

### 4.15. (±)-2-(1-Methyl-2-pyrrolidinyl)ethyl 3-bromobenzoate *(**10**)*

Prepared as described above in general procedure (Section 4.4). (±)-2-(1-methyl-2-pyrrolidinyl)ethanol (3.7 mmol, 0.5 mL) was added to a solution of 3-bromobenzoyl chloride (3.7 mmol, 0.49 mL); Yield 690 mg (60.1%); m.p. 105.3–107.1 °C; IR (cm^−1^) 2957 (stretch. C-H aliph.), 1716 (stretching COO), 1264 (stretching C-O), 1110 (stretching C-N), 748 (bending C-H Arom.); ^1^H-NMR (400 MHz, D_2_O) δ: 8.03 (s, 1H), 7.89 (d, *J* = 7.9 Hz, 1H), 7.77 (d, *J* = 8.0 Hz, 1H), 7.39 (t, *J* = 7.9 Hz, 1H), 4.45 (m, 2H), 3.72 (m, 1H), 3.50 (m, 1H), 3.19 (m, 1H), 2.98 (s, 3H), 2.45 (m, 2H), 2.12 (m, 3H), 1.86 (m, 1H); ^13^C-NMR (101 MHz, DMSO-*d*_6_) δ: 164.3, 136.2, 131.7, 131.6, 131.1, 128.3, 121.9, 65.2, 62.4, 57.6, 38.2, 29.3, 28.7, 21.0; HRMS *m/z* calcd. for C_14_H_18_BrNO_2_ (M^+^), 312.0598; found, 312.0610.

### 4.16. (±)-2-(1-Methyl-2-pyrrolidinyl)ethyl 2-methylbenzoate *(**11**)*

Prepared as described above in general procedure (Section 4.4). (±)-2-(1-methyl-2-pyrrolidinyl)ethanol (3.7 mmol, 0.5 mL) was added to a solution of 2-methylbenzoyl chloride (3.7 mmol, 0.48 mL); Yield 663 mg (73.1%); m.p. 116.1–118.0 °C; IR (cm^−1^) 2965 (stretch. C-H aliph.), 1727 (stretching COO), 1240 (stretching C-O), 1091 (stretching C-N), 740 (bending C-H Arom.); ^1^H-NMR (400 MHz, D_2_O) δ: 7.86 (d, *J* = 7.8 Hz, 1H), 7.53 (t, *J* = 7.5 Hz, 1H), 7.36 (dd, *J* = 13.0, 7.4 Hz, 2H), 4.43 (m, 2H), 3.69 (dt, *J* = 12.3, 6.7 Hz, 1H), 3.49 (ddt, *J* = 13.5, 10.1, 4.3 Hz, 1H), 3.20 (dd, *J* = 20.0, *J* = 8.5 Hz, 1H), 2.94 (s, 3H), 2.53 (s, 3H), 2.46 (m, 2H), 2.10 (m, 3H), 1.88 (m, 1H); ^13^C-NMR (101 MHz, DMSO-*d*_6_) δ: 166.6, 139.2, 132.24, 131.6, 130.2, 129.2, 126.0, 65.24, 61.7, 54.7, 40.1, 38.0, 29.2, 28.7, 21.1; HRMS *m*/*z* calcd. for C_15_H_21_NO_2_ (M^+^), 248.1648; found, 248.1650.

### 4.17. (±)-2-(1-Methyl-2-pyrrolidinyl)ethyl 4-methylbenzoate *(**12**)*

Prepared as described above in general procedure (Section 4.4). (±)-2-(1-methyl-2-pyrrolidinyl)ethanol (3.7 mmol, 0.5 mL) was added to a solution of 4-methylbenzoyl chloride (3.7 mmol, 0.49 mL); Yield 650 mg (71.4%); m.p. 174.4–175.6 °C; IR (cm^−1^) 2961 (stretch. C-H aliph.), 1721 (stretching COO), 1284 (stretching C-O), 1118 (stretching C-N), 757 (bending C-H Arom.); ^1^H-NMR (400 MHz, D_2_O) δ: 7.93 (d, *J* = 8.2 Hz, 2H), 7.38 (d, *J* = 8.0 Hz, 2H), 4.46 (m, 2H), 3.74 (m, 1H), 3.52 (m, 1H), 3.19 (dt, *J* = 8.7 Hz, 1H), 2.98 (s, 3H), 2.48 (m, 2H), 2.43 (s, 3H), 2.12 (m, 3H), 1.90 (m, 1H); ^13^C-NMR (101 MHz, DMSO-*d*_6_) δ: 165.6, 143.8, 129.3 (2C), 129.3 (2C), 126.8, 65.4, 61.7, 54.7, 38.1, 29.3, 28.8, 21.1, 21.0; HRMS *m/z* calcd. for C_15_H_21_NO_2_ (M^+^), 248.1648; found, 248.1659.

### 4.18. (S)-(1-Methyl-2-pyrrolidinyl)methyl 2-(4-(trifluoromethyl)phenyl)acetate *(**13**)*

Prepared as described above in general procedure (Section 4.3). (***S***)-(-)-(1-methyl-2-pyrrolidinyl)methanol (2.45 mmol, 0.29 mL) was added to a solution of 4-trifluoromethylphenylacetyl chloride (2.45 mmol, 0.5 gr); Yield 205 mg (27.9%); liquid brown; IR (cm^−1^) 2905 (stretch. C-H aliph.), 1740 (stretching COO), 1235 (stretching C-O), 1120 (stretching C-N), 810 (bending C-H Arom.); ^1^H-NMR (400 MHz, D_2_O) δ: 7.74 (d, *J* = 8.1 Hz, 2H), 7.53 (d, *J* = 8.0 Hz, 2H), 4.54 (dd, *J* = 12.9, *J* = 2.4 Hz, 1H), 4.34 (dd, *J* = 12.9, *J* = 6.6 Hz, 1H), 3.94 (m, 2H), 3.77 (dd, *J* = 15.6, 6.6 Hz, 1H), 3.65 (m, 1H), 3.17 (m, 1H), 2.85 (s, 3H), 2.31 (m, 1H), 2.15 (m, 1H), 1.91 (m, 2H); ^13^C-NMR (101 MHz, DMSO-*d*_6_) δ: 171.3, 135.1, 129.4, 129.1 (2C), 127.5 (2C), 124.6, 69.3, 63.3, 58.7, 54.9, 39.7, 26.1, 21.7.

### 4.19. (S)-(1-Methyl-2-pyrrolidinyl)methyl 2-(2,4-dichlorophenyl)acetate *(**14**)*

Prepared as described above in general procedure (Section 4.3). (***S***)-(-)-(1-methyl-2-pyrrolidinyl)methanol (4.2 mmol, 0.5 mL) was added to a solution of 2,4-chlorophenylacetyl chloride (4.2 mmol, 0.51 gr); Yield 339 mg (26.7%); m.p. 179.1–180.5 °C; IR (cm^−1^) 2970 (stretch. C-H aliph.), 1741 (stretching COO), 1231 (stretching C-O), 1049 (stretching C-N), 846 (bending C-H Arom.), 745 (bending C-H Arom.); ^1^H-NMR (400 MHz, D_2_O) δ: 7.55 (d, *J* = 7.7 Hz, 1H), 7.35 (m, 2H), 4.55 (dd, *J* = 13.0, *J* = 1.9 Hz, 1H), 4.36 (dd, *J* = 12.9, *J* = 6.6 Hz, 1H), 4.03 (s, 2H), 3.75 (m, 1H), 3.62 (m, 1H), 3.18 (m, 1H), 2.85 (s, 3H), 2.31 (m, 1H), 2.14 (m, 1H), 1.92 (m, 2H); ^13^C-NMR (101 MHz, DMSO-*d*_6_) δ: 171.2, 135.2, 134.1, 130.5, 129.3, 127.9, 127.0, 68.6, 64.0, 58.9, 52.1, 38.1, 28.6, 21.4; HRMS *m/z* calcd. for C_14_H_17_Cl_2_NO_2_ (M^+^), 302.0663; found, 302.0686.

### 4.20. (S)-(1-Methyl-2-pyrrolidinyl)methyl 2-(3,4-dichlorophenyl)acetate *(**15**)*

Prepared as described above in general procedure (Section 4.3). (***S***)-(-)-(1-methyl-2-pyrrolidinyl)methanol (4.2 mmol, 0.5 mL) was added to a solution of 3,4-chlorophenylacetyl chloride (4.2 mmol, 0.51 gr); Yield 339 mg (26.7%); m.p. 183.5–184.8 °C; IR (cm^−1^) 2974 (stretch. C-H aliph.), 1739 (stretching COO), 1239 (stretching C-O), 1043 (stretching C-N), 832 (bending C-H Arom.), 741 (bending C-H Arom.); ^1^H-NMR (400 MHz, D_2_O) δ: 7.48 (d, *J* = 7.2 Hz, 2H), 7.21 (d, *J* = 8.3 Hz, 1H), 4.50 (dd, *J* = 12.9, *J* = 2.5 Hz, 1H), 4.32 (dd, *J* = 12.9, *J* = 6.5 Hz, 1H), 3.80 (s, 2H), 3.71 (m, 2H), 3.20 (dt, *J* = 16.0, 8.2 Hz, 1H), 2.88 (s, 3H), 2.29 (m, 1H), 2.15 (m, 1H), 1.92 (m, 2H); ^13^C-NMR (101 MHz, DMSO-*d*_6_) δ: 170.5, 136.6, 132.1, 130.7, 130.2, 129.5, 129.1, 66.4, 63.5, 54.9, 51.4, 38.8, 28.9, 21.7; HRMS *m/z* calcd. for C_14_H_17_Cl_2_NO_2_ (M^+^), 302.0663; found, 302.0697.

### 4.21. (±)-2-(-1-Methyl-2-pyrrolidinyl)ethyl 2-phenylacetate *(**16**)*

Prepared as described above in general procedure (Section 4.5). (±)-2-(1-methyl-2-pyrrolidinyl)ethanol (4.1 mmol, 0.56 mL) was added to a solution of phenylacetyl chloride (4.1 mmol, 0.5 gr); Yield 355 mg (35.0%); m.p. 86.7–87.4 °C; IR (cm^−1^) 2961 (stretch. C-H aliph.), 1717 (stretching COO), 1280 (stretching C-O), 1115 (stretching C-N), 716 (bending C-H Arom.); ^1^H-NMR (400 MHz, D_2_O) δ 8.04 (d, *J* = 7.6 Hz, 2H), 7.72 (t, *J* = 7.5 Hz, 1H), 7.57 (t, *J* = 7.8 Hz, 2H), 4.48 (m, 2H), 3.74 (m, 2H), 3.53 (td, *J* = 5.2, 2.5 Hz, 1H), 3.45 (m, 1H), 3.18 (dt, *J* = 18.6, 9.1 Hz, 1H), 2.98 (s, 3H), 2.49 (m, 2H), 2.12 (m, 3H), 1.88 (m, 1H); ^13^C-RMN (101 MHz, DMSO-*d*_6_) δ: 171.3, 135.1, 129.5, 129.4, 129.1, 127.2, 124.6, 69.3, 63.3, 58.7, 39.5, 38.1, 29.3, 28.7, 21.0; HRMS *m/z* calcd. for C_15_H_21_NO_2_ (M^+^), 248.1648; found, 248.1667.

### 4.22. (±)-2-(-1-Methyl-2-pyrrolidinyl)ethyl 2-(3,4-dichlorophenyl)acetate *(**17**)*

Prepared as described above in general procedure (Section 4.5). (±)-2-(1-methyl-2-pyrrolidinyl)ethanol (2.4 mmol, 0.33 mL) was added to a solution of 3,4-dichlorophenylacetyl chloride (2.4 mmol, 0.5 gr); Yield 154 mg (20.0%); m.p. 148.8–150.4 °C; IR (cm^−1^) 2977 (stretch. C-H aliph.), 1732 (stretching COO), 1235 (stretching C-O), 1034 (stretching C-N), 828 (bending C-H Arom.), 736 (bending C-H Arom.); ^1^H-NMR (400 MHz, D_2_O) δ 7.41 (t, *J* = 5.5 Hz, 2H), 7.17 (dd, *J* = 8.3, *J* = 1.6 Hz, 1H), 4.21 (m, 2H), 3.68 (d, *J* = 6.6 Hz, 3H), 3.23 (m, 1H), 3.10 (d, *J* = 9.9 Hz, 1H), 2.86 (s, 3H), 2.30 (m, 1H), 2.20 (m, 1H), 2.03 (m, 2H), 1.90 (m, 1H), 1.71 (m, 1H); ^13^C-NMR (101 MHz, DMSO-*d*_6_) δ: 170.2, 135.2, 131.4, 130.5, 130.2, 129.9, 129.4, 65.1, 61.4, 54.5, 38.6, 37.9, 29.0, 28.4, 20.8; HRMS *m/z* calcd. for C_15_H_19_Cl_2_NO_2_ (M^+^), 316.0868; found, 316.0896.

### 4.23. (S)-(1-Methyl-2-pyrrolidinyl)methyl 2-naphthoate *(**18**)*

Prepared as described above in general procedure (Section 4.2). (***S***)-(-)-(1-methyl-2-pyrrolidinyl)methanol (4.2 mmol, 0.5 mL) was added to a solution of 2-naphthoyl chloride (4.2 mmol, 0.80 gr); Yield 735 mg (65%); m.p. 125.5–127.1 °C; IR (cm^−1^) 3057 (stretch. C-H aliph.), 1748 (stretching COO), 1260 (stretching C-O), 1123 (stretching C-N), 757 (bending C-H Arom.); ^1^H-NMR (400 MHz, DMSO-*d*_6_) δ: 8.61 (s, 1H), 8.16 (m, 1H), 8.01 (m, 3H), 7.65 (dt, *J* = 13.61, 6.68, 6.68 Hz, 2H), 4.69 (m, 2H), 3.85 (m, 1H), 3.03 (m, 2H), 2.83 (s, 3H), 2.07 (m, 1H), 1.96 (m, 1H), 1.85 (m, 1H), 1.72 (m, 1H); ^13^C-NMR (101 MHz, DMSO-*d*_6_) δ: 167.4, 134.9, 132.2, 130.5, 129.3, 128.3, 128.2, 128.1, 127.7, 126.8, 125.2, 69.4, 59.1, 55.9, 39.5, 26.0, 21.8; HRMS *m/z* calcd. for C_17_H_19_NO_2_ (M^+^), 270.1498; found, 270.1517.

### 4.24. (±)-2-(1-Methyl-2-pyrrolidinyl)ethyl 2-naphthoate *(**19**)*

Prepared as described above in general procedure (Section 4.4). (±)-2-(1-methyl-2-pyrrolidinyl)ethanol (3.7 mmol, 0.5 mL) was added to a solution of 2-naphthoyl chloride (3.7 mmol, 0.7 gr); Yield 637 mg (61.1%); m.p. 155.6–157.4 °C; IR (cm^−1^) 2948 (stretch. C-H aliph.), 1727 (stretching COO), 1288 (stretching C-O), 1200 (stretching C-N), 781 (bending C-H Arom.); ^1^H-NMR (400 MHz, D_2_O) δ: 8.67 (s, 1H), 8.16 (d, *J* = 7.71 Hz, 1H), 8.03 (m, 3H), 7.66 (dt, *J* = 21.77, 7.17, 7.17 Hz, 2H), 4.45 (m, 2H), 3.55 (dt, *J* = 13.54, 6.59, 6.59 Hz, 1H), 3.46 (m, 1H), 3.01 (m, 1H), 2.82 (s, 3H), 2.33 (m, 1H), 2.18 (m, 1H), 1.99 (m, 3H), 1.79 (m, 1H); ^13^C-NMR (101 MHz, DMSO-*d*_6_) δ: 165.7, 135.1, 132.1, 130.7, 129.4, 128.7, 128.4, 127.7, 127.0, 126.8, 124.8, 65.4, 62.1, 57.6, 38.1, 29.3, 28.8, 21.0; HRMS *m/z* calcd. for C_18_H_21_NO_2_ (M^+^), 284.1648; found, 284.1677.

### 4.25. (S)-(1-Methyl-2-pyrrolidinyl)methyl 2,2-diphenylacetate *(**20**)*

Prepared as described above in general procedure (Section 4.3). (***S***)-(-)-(1-methyl-2-pyrrolidinyl)methanol (2.2 mmol, 0.26 mL) was added to a solution of 2,2-diphenylacetyl chloride (2.2 mmol, 0.5 gr); Yield 293 mg (43.1%); m.p. 121.5–122.3 °C; IR (cm^−1^) 2991 (stretch. C-H aliph.), 1742 (stretching COO), 1200 (stretching C-O), 1063 (stretching C-N), 827 (bending C-H Arom.), 736 (bending C-H Arom.); ^1^H-NMR (400 MHz, D_2_O) δ: 7.39 (m, 10H), 5.34 (s, 1H), 4.54 (dd, *J* = 12.9, *J* = 2.6 Hz, 1H), 4.36 (dd, *J* = 12.9, *J* = 7.1 Hz, 1H), 3.62 (td, *J* = 10.4, 8.8, 4.2 Hz, 1H), 3.52 (dt, *J* = 11.9, 6.2 Hz, 1H), 3.06 (dt, *J* = 11.2, 8.1 Hz, 1H), 2.64 (s, 3H), 2.23 (m, 1H), 2.05 (m, 1H), 1.84 (m, 1H), 1.73 (m, 1H); ^13^C-NMR (101 MHz, DMSO-*d*_6_) δ: 171.9, 139.0 (2C), 129.5 (4C), 129.3 (4C), 127.5 (2C), 69.9, 63.7, 54.2, 52.8, 37.2, 26.8, 20.5; HRMS *m/z* calcd. for C_20_H_23_NO_2_ (M^+^), 310.1761; found, 310.1795.

### 4.26. (±)-2-(-1-Methyl-2-pyrrolidinyl)ethyl 2,2-diphenylacetate *(**21**)*

Prepared as described above in general procedure (Section 4.4). (±)-2-(1-methyl-2-pyrrolidinyl)ethanol (2.2 mmol, 0.3 mL) was added to a solution of 2,2-diphenylacetyl chloride (2.17 mmol, 0.5 gr); Yield 217 mg (31.0%); m.p. 115.3–117.2 °C; IR (cm^−1^) 2984 (stretch. C-H aliph.), 1735 (stretching COO), 1195 (stretching C-O), 1151 (stretching C-N), 744 (bending C-H Arom.), 700 (bending C-H Arom.); ^1^H-NMR (400 MHz, D_2_O) δ: 7.28 (m, 10H), 5.16 (s, 1H), 4.23 (m, 2H), 3.57 (dt, *J* = 12.5, 6.6 Hz, 1H), 2.93 (m, 1H), 2.86 (m, 1H), 2.65 (s, 3H), 2.19 (m, 1H), 1.98 (m, 1H), 1.88 (m, 2H), 1.79 (m, 1H), 1.53 (dq, *J* = 12.4, 8.5 Hz, 1H); ^13^C-NMR (101 MHz, DMSO-*d*_6_) δ: 171.6, 138.6 (2C), 128.3 (8C), 126.9 (2C), 65.0, 61.7, 55.5, 54.4, 37.8, 28.9, 28.3, 20.7; HRMS *m/z* calcd. for C_21_H_25_NO_2_ (M^+^), 324.1968; found, 324.1998.

## 5. Conclusions

Radioligand binding assays supported with docking studies were carried out with a series of novel pyrrolidine ester derivatives in α4β2 nAChR, h-DAT and h-SERT. The binding assays performed on h-SERT cells gave no affinity for any synthesized compounds in the central site of h-SERT [49]. However, compounds **2**, **4**, **8**, **10**, **18** and **19** produced an increase in the binding of [^3^H]-paroxetine, which may be interpreted as an allosteric h-SERT modulation. This anomalous behavior could be assigned to a steric hindrance effect on the radioligand, induced by an ionic bond and hydrophobic interactions among our compounds with Glu 494 and Phe 335 residues in the allosteric binding site of h-SERT, avoiding the radioligand leaving of the central site [49]. This is interesting since suggest that the aforementioned compounds could increase the efficacy of SSRIs if used in conjunction. 

On the other hand, compound **20** displays the desired profile, positioning it as the unique compound exhibiting triple affinity of the series. In SERT assays, compound **20** induced the highest [^3^H]-paroxetine binding, a result supported by docking studies which revealed the interaction of both phenyl groups with Phe 335 and Phe 556 residues on the allosteric binding site of h-SERT. In the same context, compound **20** showed h-DAT interaction between one of the phenyl groups with Tyr 548 at the central binding site, reinforcing its affinity. Docking studies of compound **20** at α4β2 nAChR, showed that one of the phenyl group adopted a conformation fitting the space between Phe 119 and Trp 182 residues. On the other hand, the high h-DAT affinity obtained for compound **21** can be explained considering the interactions unveiled by docking studies. In this case, both phenyl groups of **21** interacted with Tyr 470 residue. Regarding α4β2 nAChR affinity, the π–cation interactions with Tyr 230 residue and the hydrophobic interaction with Phe 119 residue probably favored the affinity. Based on these results, the presence of two geminal phenyl groups at the alfa-position in ester derivatives resulted fundamental to obtain compounds with enhanced affinity.

Finally, this extended study conducted to promissory results in the design and finding of novel multi-ligands simultaneously acting at different targets.

The rational design of multi-target compounds is far from being an easy task, dealing with the crucial issues of selecting the right target combination, achieving a balanced activity towards them, and excluding activity at the undesired target(s), while at the same time retaining drug-like properties. 

## Figures and Tables

**Figure 1 molecules-24-03808-f001:**
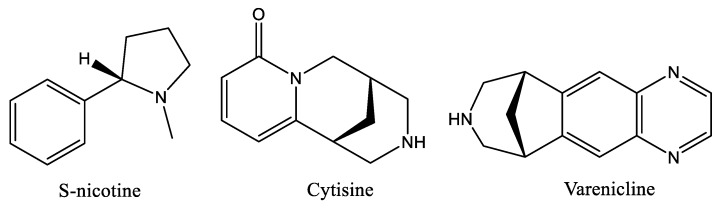
Chemical structure of alkaloids S-nicotine, Cytisine and smoking cessation drug Varenicline.

**Figure 2 molecules-24-03808-f002:**
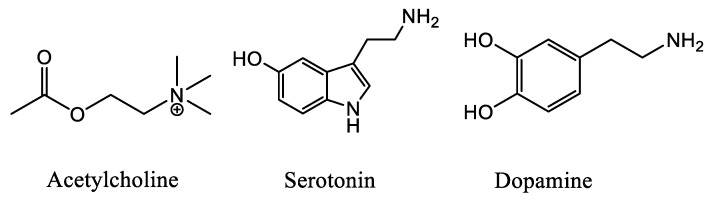
Chemical structure of the neurotransmitters ACh, 5-HT and DA.

**Figure 3 molecules-24-03808-f003:**
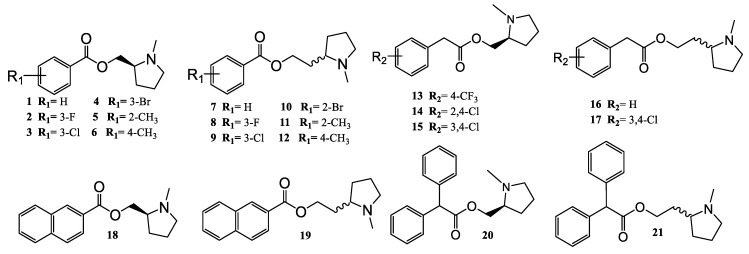
Structures of the 21 synthesized compounds used in this study.

**Figure 4 molecules-24-03808-f004:**
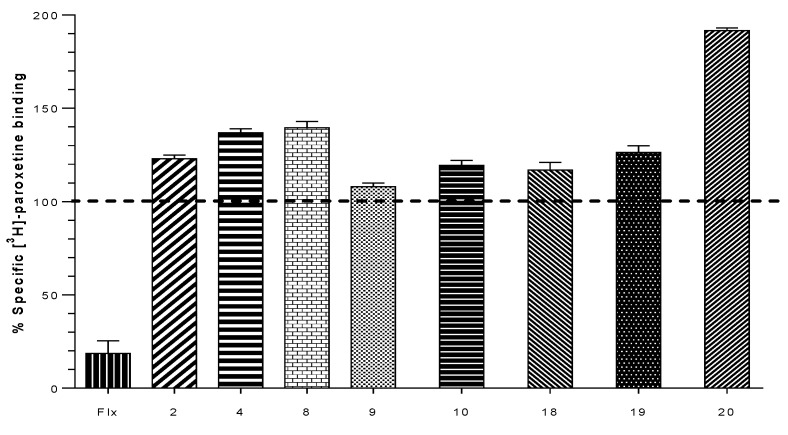
Graph obtained for the displacement experiments of [^3^H]-paroxetine for the synthesized compounds [1,2,3,4,5,6,7,8,9,10,11,12,13,14,15,16,17,18,19,20,21] in the cellular background h-SERT induced to 2.5 × 10^−5^ M concentration. Fluoxetine (Flx) was used as a control for the compounds under study.

**Figure 5 molecules-24-03808-f005:**
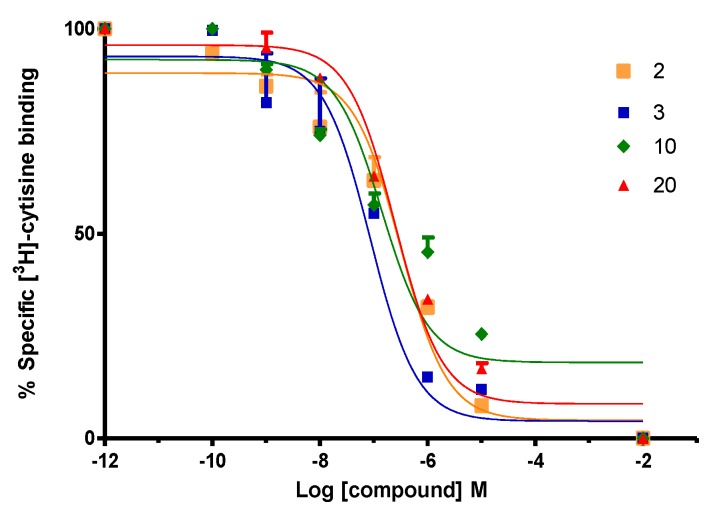
Compounds effects on binding of [^3^H]-cytisine to α4β2 nAChRs. Data points represent the mean ± SEM of four experiments, each one carried out in triplicate. The radioligand concentration in all displacement studies was 1 nM.

**Figure 6 molecules-24-03808-f006:**
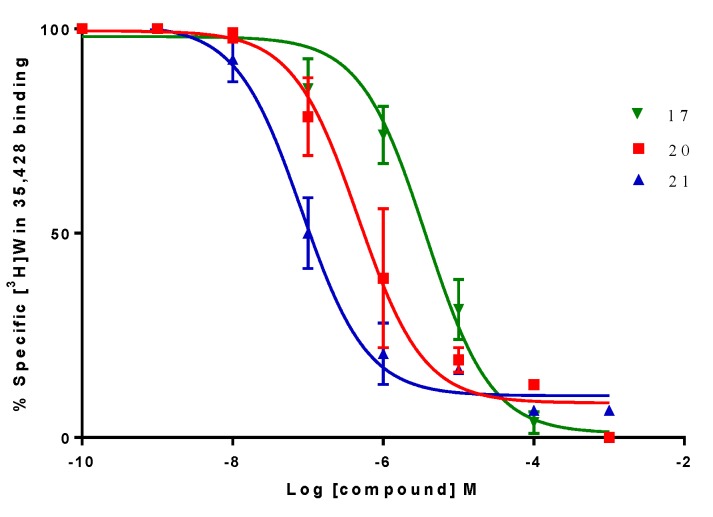
Graph obtained for the displacement experiment of [^3^H]-WIN 35428. Data points represent the mean ± SEM of four experiments, each one carried out in triplicate. The radioligand concentration in all displacement studies was 1 nM.

**Figure 7 molecules-24-03808-f007:**
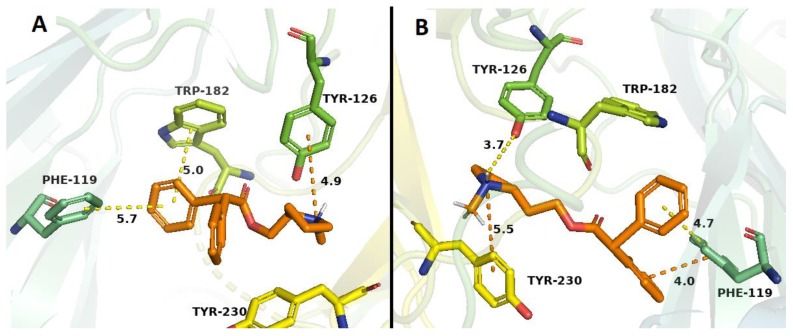
Binding modes of compound **20** (**A**) and compound **21** (**B**) to the human α4β2 nAChR. The orange-ligands denotes compounds **20** and **21.** Dotted lines correspond to the different interactions which may take place in the ligand–receptor complexes.

**Figure 8 molecules-24-03808-f008:**
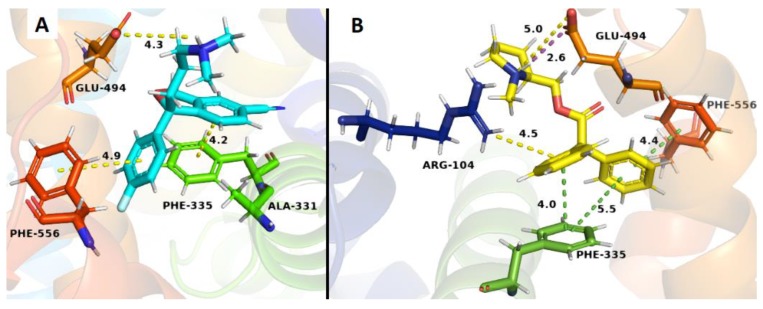
Binding modes of S-citalopram co-crystal (**A**) and compound **20** (**B**) to the human SERT. The cyan ligand denotes S-citalopram and the yellow ligand denotes compound **20.** Dotted lines correspond to the different interactions in the ligand–receptor complexes.

**Figure 9 molecules-24-03808-f009:**
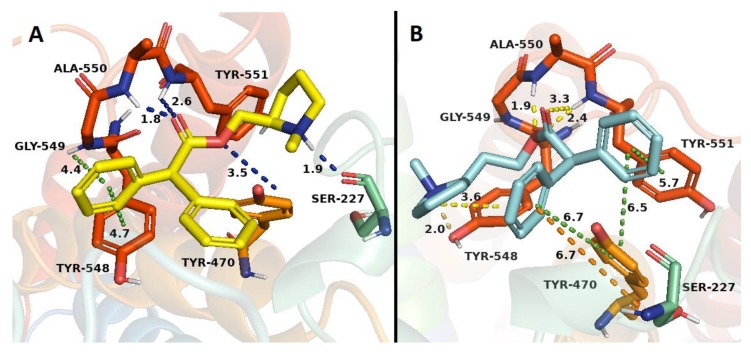
Binding modes of compounds **20** (**A**) and **21** (**B**) to the human DAT. The yellow ligand denotes compound **20** and the cyan ligand corresponds to compound **21**. Dotted lines correspond to the different interactions taking place in the ligand-receptor complexes.

**Table 1 molecules-24-03808-t001:** K_i_ values (μM) for the synthesized compounds against h-DAT, α4β2 nAChRs and [^3^H]-paroxetine h-SERT expressed as binding percentage.

Compound	% of Binding [^3^H]-Paroxetine h-SERT	K_i_ (µM) h-DAT	K_i_ (µM) α4β2 nAChR
**Bupropion**	N.E	0.370	10.0
**1**	86.0 ± 0.6	N.E	0.023 ± 0.006
**2**	123.0 ± 1.2	N.E	0.094 ± 0.002
**3**	87.5 ± 1.5	N.E	0.009 ± 0.001
**4**	137.0 ± 1.2	N.E	0.132 ± 0.038
**7**	100.0 ± 0.5	N.E	1.788 ± 0.378
**8**	139.5 ± 2.0	N.E	N.E
**9**	108.0 ± 1.2	N.E	3.461 ± 0.360
**10**	119.3 ± 2.7	N.E	0.042 ± 0.004
**16**	92.0 ± 0.3	22.690 ± 7.099	N.E
**17**	78.0 ± 0.6	3.317 ± 0.923	N.E
**18**	117.0 ± 2.3	99.330 ± 1.411	0.120 ± 0.037
**19**	126.3 ± 1.9	44.240 ± 8.120	N.E
**20**	191.5 ± 0.9	1.208 ± 0.230	0.023 ± 0.006
**21**	86.5 ± 1.5	0.075 ± 0.009	0.113 ± 0.037

The non-specific binding of [^3^H]-paroxetine (2 nM) in h-SERT exhibited a radioligand displacement in the presence or absence of 25 μM of the compound under study. Fluoxetine (Flx, 25 mM) was used to define the nonspecific ligand. The non-specific binding in the h-DAT exhibited a radioligand displacement of [^3^H]-WIN 35,428 at 1 nM concentration the dissociation constant (K_d_) used to estimate K_i_ was 9.21 nM for [^3^H]-WIN 35428. In the α4β2 nAChR displacement assays the concentration of the radio ligand was 1 nM [^3^H]-Cytisine and its K_d_ = 0.43 nM used for the K_i_ calculation. The bupropion affinity was just included as a comparative value and was not used in the trial. N.E = no effect at 100 µM.

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
