# Peer review of "Synthesis of Novel Nicotinic Ligands with Multimodal Action: Targeting Acetylcholine α4β2, Dopamine and Serotonin Transporters"

_molecules, 2019, doi:10.3390/molecules24203808_

Round 1

Reviewer 1 Report

ID: molecules-603292

The study by González-Gutiérrez et al., entitled “Synthesis of novel nicotinic ligands with multimodal action: Targeting acetylcholine α4β2, dopamine and serotonin transporters” proposes a series of experiments consisting in chemical synthesis and pharmacological characterization, aiming at proposing new ligands having a simultaneous affinity for three types of targets: the nicotinic acetylcholine receptors, the dopamine and serotonin transporters. This multi-target strategy aims to create new drugs for the pharmacological management of neurological and/or psychiatric pathologies. The authors have implemented techniques of synthesis, binding assay and molecular modeling to emerge new candidate molecules to develop.

The introduction of this article is too light: the problematic in physiopathological and pharmacological terms is too quickly presented. The authors should detail the physiological basis of cholinergic and mono-aminergic networks as a target for psychiatric (depression, anxiety) or neurological (AD, addiction, PD) disorders. In doing so, they will have the benefit of having multi-target therapeutic ligands, that is to say having the capacity to play a role of nAChRs antagonist and inhibitors of the reuptake of DA and 5-HT. Finally, they will discuss the problems posed by such a strategy, in terms of adverse effects, because obviously, there are many with such a pharmacological approach. Their introduction must therefore be significantly restructured.

In the same way, the "Results and Discussion" part is only part of the description of the results, and absolutely not a discussion. Authors must rewrite this part by inserting a real discussion, notably by comparing their data with those published by others, and by extending it with the possible therapeutic implications of their data, which is not done in their article. Finally, the Conclusion is rather a summary of the data generated, in a non-synthetic way. It seems to me essential to open the perspectives of this work, which has been totally neglected in the article in its current form.

It is therefore little to say that in the present form, this article cannot be published, but that it must undergo a real restructuring in order to be able to consider being published. The authors must therefore work to rewrite their draft by improving it very significantly.

The draft contains a lot of English mistakes, or inaccuracies that need to be corrected.

The article contains 13 figures, making it a pretty dense draft. I think that Figures 4, 5, 6 and 7 - which are figures detailing the synthesis of the 21 compounds - could be supplementary figures, to avoid weighing the whole.

L86: Have benzoate derivatives been tested on other nAChRs than the α4β2 subtype, In particular the α7 and α3β4 subtypes? How to target a type of nAChRs (α4β2) and SERT/DAT, without linking other nAChRs ? How do you propose to solve this major pharmacological issue?

Minor

L27: associated to different health conditions, such as depressive, anxiety and addiction disorders

L50: of one or more of these systems

L58: Figure 1. Structure of the alkaloid (S)-nicotine and the neurotransmitters ACh, 5-HT and DA.

L58 & 60: ACh, not Ach (as in L44)

L73-74 should be more precise: “…which act both as inhibitors of SERT/DAT and nAChR antagonists”.

L77: Structure of classical antidepressants

L79-80: such a promiscuous profile combination structural aspect. Please reformulate

L102: Figure 3.- Structures of the 21 synthesized compounds used in this study.

L131-132: for the corresponding monoamine transporters respectively

L161-162: N.E= no effect at 100 M ?? Please correct.

L163: all compounds

L171: Figure 8 does not have a proper title. Please add one.

L178: the highest affinity ? Please precise. If so, please indicate if the difference is significant.

L194: cytisine to α4β2 nAChRs

L214: why was modafinil used here? Is it used as a DAT ligand? If so, it should be precised that modafinil mode of action is not clear.

L227: Binding studies with α4β2 nAChRs

L227: indicate that only

L228: at the ACh binding site, i.e the orthosteric site,

L230; Do you mean “nicotin-derived synthetic compounds” ?

L242:     Figure 11.- Binding modes of compounds 20 and 21 to the human α4β2 nAChR.

Please modify the title of Figure 12 accordingly.

Author Response

Dear Editor:  Please receive point by point the corresponding answers for referees’s requirements and suggestions.

Mayor corrections.

Reviewer 1 Comments and Suggestions: The introduction of this article is too light: the problematic in physiopathological and pharmacological terms is too quickly presented. The authors should detail the physiological basis of cholinergic and mono-aminergic networks as a target for psychiatric (depression, anxiety) or neurological (AD, addiction, PD) disorders. In doing so, they will have the benefit of having multi-target therapeutic ligands, that is to say having the capacity to play a role of nAChRs antagonist and inhibitors of the reuptake of DA and 5-HT. Finally, they will discuss the problems posed by such a strategy, in terms of adverse effects, because obviously, there are many with such a pharmacological approach. Their introduction must therefore be significantly restructured.

Author's response: The authors have restructured the introduction completely according to the observations and suggestions indicated by the reviewer.

Reviewer 1 Comments and Suggestions: In the same way, the "Results and Discussion" part is only part of the description of the results, and absolutely not a discussion. Authors must rewrite this part by inserting a real discussion, notably by comparing their data with those published by others, and by extending it with the possible therapeutic implications of their data, which is not done in their article. Finally, the Conclusion is rather a summary of the data generated, in a non-synthetic way. It seems to me essential to open the perspectives of this work, which has been totally neglected in the article in its current form.

Author's response: The authors changed the discussion including a stronger analysis of the results and not just a summary of the data, in the same way the conclusion was reformulated.

Reviewer 1 Comments and Suggestions: The article contains 13 figures, making it a pretty dense draft. I think that Figures 4, 5, 6 and 7 - which are figures detailing the synthesis of the 21 compounds - could be supplementary figures, to avoid weighing the whole.

Author's response: The authors removed figures 4, 5, 6 and 7, and were added as complementary material as was required by the referee 1.

Reviewer 1 Comments and Suggestions: L86: Have benzoate derivatives been tested on other nAChRs than the α4β2 subtype, In particular the α7 and α3β4 subtypes? How to target a type of nAChRs (α4β2) and SERT/DAT, without linking other nAChRs? How do you propose to solve this major pharmacological issue?

Author's response: In this manuscript, the authors have not carried out studies of benzoate derivatives in nAChR α3β4 but is not ruled out to do so in future studies. However, previous studies on affinity for nAChR α7 shown that they bind with similar affinity to the nAChR α4β2.  These results were not included here given that our primarily goals were to find novel structures acting as multitarget agents. (Faundez-Parraguez M.; et al. Neonicotinic analogues: Selective antagonists for α4β2 nicotinic acetylcholine receptors. Bioorg. Med. Chem. 2013, 21, 2687–94. DOI: 10.1016/j.bmc.2013.03.024).

Minor corrections.

Reviewer 1 Comments and Suggestions:

L27: associated to different health conditions, such as depressive, anxiety and addiction disorders

L50: of one or more of these systems

L58: Figure 1. Structure of the alkaloid (S)-nicotine and the neurotransmitters ACh, 5-HT and DA.

L58 & 60: ACh, not Ach (as in L44)

L73-74 should be more precise: “…which act both as inhibitors of SERT/DAT and nAChR antagonists”.

L77: Structure of classical antidepressants

L79-80: such a promiscuous profile combination structural aspect. Please reformulate

L102: Figure 3.- Structures of the 21 synthesized compounds used in this study.

L131-132: for the corresponding monoamine transporters respectively

L161-162: N.E= no effect at 100 M ?? Please correct.

L163: all compounds

L171: Figure 8 does not have a proper title. Please add one.

L178: the highest affinity? Please precise. If so, please indicate if the difference is significant.

L194: cytisine to α4β2 nAChRs

L214: why was modafinil used here? Is it used as a DAT ligand? If so, it should be precised that modafinil mode of action is not clear.

L227: Binding studies with α4β2 nAChRs

L227: indicate that only

L228: at the ACh binding site, i.e the orthosteric site,

L230; Do you mean “nicotin-derived synthetic compounds”?

L242: Figure 11.- Binding modes of compounds 20 and 21 to the human α4β2 nAChR. Please modify the title of Figure 12 accordingly.

Author's response: The authors made all minor corrections proposed by the reviewer.

Reviewer 2 Report

The present communication describes novel nicotinic ligands targeting acetylcholine α4β2, dopamine and 3 serotonin transporters, and is a continuation of a previous report by the same group (ref 9) on the same biological targets and related molecules. However, the results are of interest, and consequently I suggest to accept it after revision as follows:

Please comment of the functional assay (agonist/antagonist) profile of compound 20 and 21. Correct everywhere in the text: compound xx, as compound (not as Bold) xx. The Ki values for compounds 20 and 21 in the Abstract are not those shown in Table 1. Please correct… 20 and 21 in the Abstract should be in bold. Please correct.

Author Response

Dear Editor:  Please receive point by point the corresponding answers for referees’s requirements and suggestions.

Reviewer 2 Comments and Suggestions: Please comment of the functional assay (agonist/antagonist) profile of compound 20 and 21.

Author's response: The authors did not perform functional trials in this work.

Reviewer 2 Comments and Suggestions: Correct everywhere in the text: compound xx, as compound (not as Bold) xx.

Author's response: The authors have corrected the text suggested by the reviewer.

Reviewer 2 Comments and Suggestions: The Ki values for compounds 20 and 21 in the Abstract are not those shown in Table 1.

Author's response: The authors have corrected Ki values in the abstract correlating now with the data shown in Table 1.

Reviewer 2 Comments and Suggestions: 20 and 21 in the Abstract should be in bold.

Author's response: The authors have corrected 20 and 21 in the abstract.

Round 2

Reviewer 1 Report

The authors have made substantial changes to their draft and have, from my point of view, significantly improved. There are still minor changes in the English language expression, but I think that can be improved during the editing process.